# Uniaxial-strain control of nematic superconductivity in $Sr_xBi_2Se_3$

Ivan Kostylev [1✉], Shingo Yonezawa [1✉], Zhiwei Wang[2,3], Yoichi Ando [2] & Yoshiteru Maeno[1]

Nematic states are characterized by rotational symmetry breaking without translational ordering. Recently, nematic superconductivity, in which the superconducting gap spontaneously lifts the rotational symmetry of the lattice, has been discovered. In nematic superconductivity, multiple superconducting domains with different nematic orientations can exist, and these domains can be controlled by a conjugate external stimulus. Domain engineering is quite common in magnets but has not been achieved in superconductors. Here, we report control of the nematic superconductivity and their domains of $Sr_xBi_2Se_3$, through externally-applied uniaxial stress. The suppression of subdomains indicates that it is the $\Delta_{4y}$ state that is most favoured under compression along the basal Bi-Bi bonds. This fact allows us to determine the coupling parameter between the nematicity and lattice distortion. These results provide an inevitable step towards microscopic understanding and future utilization of the unique topological nematic superconductivity.

[1] Department of Physics, Graduate School of Science, Kyoto University, Kyoto 606-8502, Japan. [2] Institute of Physics II, University of Cologne, Köln 50937, Germany. [3] Key Laboratory of Advanced Optoelectronic Quantum Architecture and Measurement, Ministry of Education (MOE), School of Physics, Beijing Institute of Technology, Beijing 100081, P. R. China. ✉email: kostylev@scphys.kyoto-u.ac.jp; yonezawa@scphys.kyoto-u.ac.jp

In nematic states of liquid crystals, bar-shaped molecules exhibit orientational ordering. Because of the peculiar partial ordering property, the orientation of the molecules and hence the structure around defects are easily controlled by external stimuli, as widely utilized in liquid-crystal displays. Analogous phenomena in electronic systems, nematic electron liquids, where conduction electrons exhibit orientational ordering, have been revealed[1,2]. Here, orientational properties are also highly controllable, and observations of such tunability have played fundamental roles to clarify driving mechanisms[3,4].

A more exotic form of nematicity has been discovered in $A_x Bi_2 Se_3$ ($A = $ Cu, Sr, Nb)[5–9]: nematic superconductivity[5], in which the superconducting (SC) gap amplitude spontaneously lifts the rotational symmetry of the lattice as a consequence of Cooper-pair formation. After the initial observations, nematic SC behavior has been consistently observed by using various probes such as transport, thermodynamic, and microscopic techniques[10–18], as reviewed in ref. [19]. We comment here that these doped $Bi_2 Se_3$ nematic superconductors are quite distinct from iron pnictide superconductors, where nematicity occurs in the normal state as a consequence of orbital ordering and superconductivity eventually occurs at much lower temperatures[4]. Although control of such normal-state nematicity has been intensively tested in iron pnictides, a direct control of nematicity of Cooper pairs has never been achieved. It is essential to show such control over the nematic SC orientation for further research developments of nematic superconductivity.

In this Article, we report the first control of nematic superconductivity in $Sr_x Bi_2 Se_3$, through application of in-situ tunable uniaxial stress along the $a$ axis (meaning a Bi−Bi bond direction). We reversibly controlled the nematic domain structure, allowing us to determine the sign of the coupling constant between the nematicity and lattice distortion.

## Results

**Doped $Bi_2 Se_3$ nematic superconductors**. Our target materials family $A_x Bi_2 Se_3$ is derived from the topological insulator $Bi_2 Se_3$[20,21], which has a trigonal crystalline symmetry with three equivalent crystalline $a$ axes in the basal plane (Fig. 1a)[22]. Because the superconductivity induced by $A$ ion intercalation[23–25] occurs in its topologically non-trivial bands[26,27], the resultant superconductivity can also be topologically non-trivial. Indeed, topological SC states have been proposed, among which a pair of SC states in the two-dimensional $E_u$ representation, $\Delta_{4x}$ and $\Delta_{4y}$, are nematic SC states[5,28–30]. The SC gap amplitude of the $\Delta_{4x}$ and $\Delta_{4y}$ states are twofold anisotropic and their maximum amplitude is located along the $a$ and $a^*$ axes, respectively (Fig. 1a). Since there are three equivalent basal axes, each state has three degenerate order-parameter orientations, as shown in Fig. 2c. In addition, it has been shown that there is sample-to-sample variation in whether the nematicity aligns along the $a$ or $a^*$ axes [11,16]. This fact suggests that $\Delta_{4x}$ and $\Delta_{4y}$ states are nearly degenerate states. Therefore, in total, a $A_x Bi_2 Se_3$ sample can contain up to six kinds of nematic SC domains, in which nematicity of each domain is selected by a certain pre-existing local symmetry breaking field such as possible structural distortion or $A$ ion distribution. Here, we probe whether applied uniaxial stress can overcome this pinning and alter the SC domain configuration.

**Magnetoresistance without and with uniaxial strain**. In this work, we measured the magnetoresistance of single-crystalline $Sr_{0.06} Bi_2 Se_3$ samples (with the critical temperature $T_c$ of 2.8 K; see Supplementary Note 1). The sample was affixed onto a custom-made uniaxial strain cell[31], a modified version of the recent invention[32], mounted inside a vector magnet. The sample was cut

along one of the $a$ axes (Bi-Bi bond direction), which we define as the $x$ axis (Figs. 1a and b). Both the uniaxial force and electric current were applied along this $x$ axis. The angle between the magnetic field and $x$ axis is denoted as $\phi_{ab}$. We comment that twofold behavior in the normal state is practically absent in our sample, and is not altered by the application of compressive strain (Supplementary Note 13). Following the ordinary convention, we defined the sign of the strain so that negative sign corresponds to compressive strain.

In Figs. 1c and 2a, b, we present the magnetoresistance at 2.2 K for various $\phi_{ab}$. First, focus on the data with the relative strain $\Delta\varepsilon_{xx}$ of 0%, i.e. zero applied voltage to the piezo stacks (black curves in Fig. 1c), corresponding to the actual strain of around $+0.10\%$ (tensile) due to the thermal-contraction difference of the sample and strain cell ("Methods" section). Clearly, superconductivity is more stable for $\phi_{ab} = -90°$ ($H \| - y$) than $0°$ ($H\|x$), resulting in a prominent twofold upper critical field $H_{c2}$, which is indicative of the nematic superconductivity[8,11]. This observed anisotropy $H_{c2}\| - y > H_{c2}\|x$ is consistent with the $\Delta_{4y}$ state with the SC gap larger along $y$[33], which is schematically shown as the $Y_0$ state in Fig. 2c. Interestingly, additional sixfold behavior emerges at the onset of the SC transition between 1 and 2 T, as clearly visible by the green region extending along $\phi_{ab} = -30°$ or $+30°$ in Fig. 2a (see Supplementary Fig. 2 for raw data). This sixfold component indicates that the sample contains minor parts exhibiting large $H_{c2}$ along $\phi_{ab} = \pm30°$, namely the $Y_1$ and $Y_2$ domains (both belonging to the $\Delta_{4y}$ state) in Fig. 2c with their gap maxima along the $\pm30°$ directions.

Next, let us focus on the data under applied strain of $\Delta\varepsilon_{xx} = -1.19\%$ (green curves in Fig. 1c) corresponding to the actual compressive strain of around $\varepsilon_{xx} \simeq -1.1\%$; the largest measured compressive strain in the elastic limit (see Supplementary Note 3). Notably, the magnetoresistance at the SC transition is substantially altered, marking the first in-situ uniaxial-strain control of nematic superconductivity. More specifically, the SC transition becomes sharper with strain except near $\phi_{ab} = -90°$ ($H\|y$). Moreover, comparing the color plots in Figs. 2a and b, we can notice that the weak sixfold SC onset due to domains, seen in the $\Delta\varepsilon_{xx} = 0$ data, is substantially reduced by the applied strain. Thus the primary effect of the compressive uniaxial strain is to suppress the minor nematic domains.

**Upper critical fields**. From the magnetoresistance data, we defined $H_{c2}$ as the field where the resistance $R(H)$ divided by the normal-state resistance $R_n$ reaches various criterion values ("Methods" section; Supplementary Note 4). In the strain dependence of $H_{c2}$ (Fig. 3), there is a high reproducibility among measurement cycles within the present strain range, manifesting that strain response is repeatable and thus our sample is in the elastic deformation regime. Reproducibility across samples has also been demonstrated (see Supplementary Note 5). Comparing data for various field directions, we can see that $H_{c2}\|x$ largely reduces under strain, attributable to the disappearance of minor nematic SC domains. In contrast, $H_{c2}$ along the $y$ and $z$ axes (Fig. 3), as well as the zero-field $T_c$ (Supplementary Fig. 7), is only weakly affected by strain, with small decreasing trend under compression.

The strain control of the nematic subdomains is more evident in the $H_{c2}(\phi_{ab})$ curves in Fig. 4a. Notice that $H_{c2}$ defined with higher values of $R/R_n$ is more sensitive to existence of nematic subdomains. In addition to the prominent twofold anisotropy with maxima at $\phi_{ab} = \pm90°$ (originating from the $Y_0$ domain) seen in all criteria, $H_{c2}$ with the 95% or 80% criteria exhibit additional 4 peaks located at $\phi_{ab} = \pm30°$ and $\pm150°$ for low $\Delta\varepsilon_{xx}$, due to the existence of $Y_1$ and $Y_2$ domains. These peaks are suppressed with

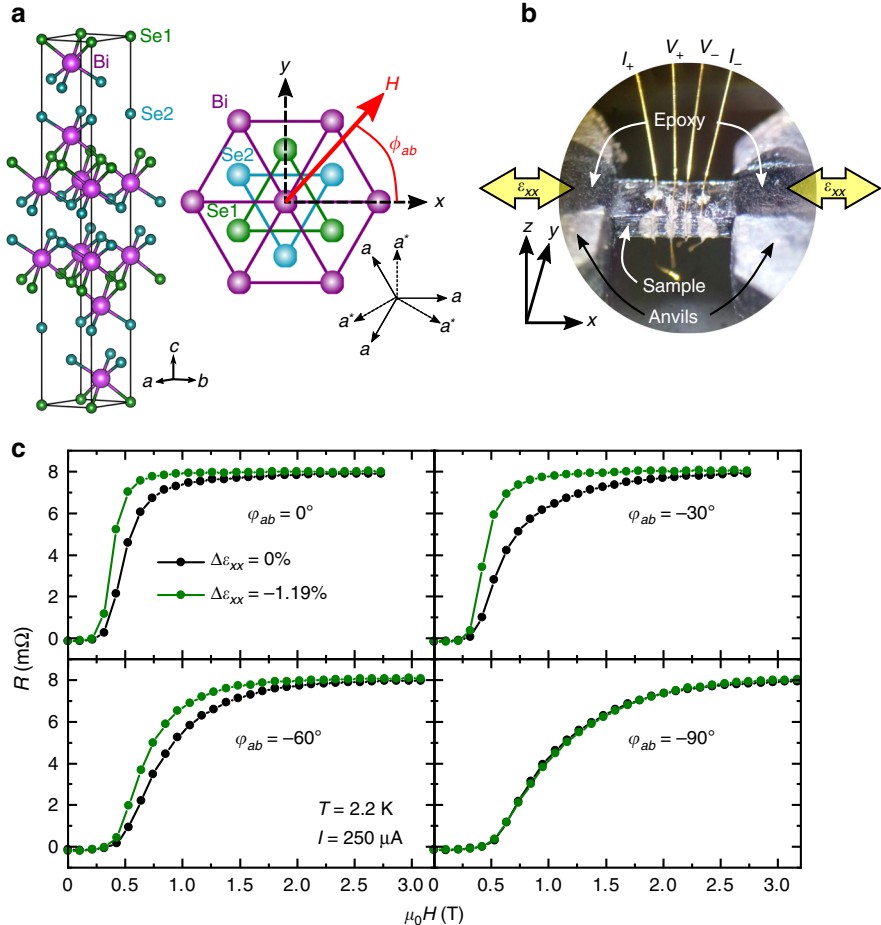

**Fig. 1 Uniaxial-strain control of nematic superconductivity in Sr$_x$Bi$_2$Se$_3$. a** Crystal structure of the mother compound Bi$_2$Se$_3$. The right figure shows the definitions of the axes and the field angle $\phi_{ab}$ with respect to the crystal structure in the *ab* plane, with three equivalent *a* and *a*$^\star$ axes. **b** Photograph of the sample in the uniaxial strain cell with 4-wire terminal configuration. *I* and *V* labels next to the gold wires indicate the current and voltage leads, respectively. The large yellow arrows indicate the direction of the external strain, which was applied parallel to the *x* axis. **c** Magnetoresistance at specified field directions in the *ab* plane ($\phi_{ab} = 0°$, $-30°$, $-60°$, $-90°$), with and without $\Delta\varepsilon_{xx}$. The data were obtained at 2.2 K and with 250 $\mu$A applied current. A substantial change in the magnetoresistance curves under large $\Delta\varepsilon_{xx}$ (green curves) provides evidence for the in-situ uniaxial-strain control of the nematic superconductivity. Source data are provided as a Source Data file.

increasing $\Delta\varepsilon_{xx}$, indicating the disappearance of the minor $Y_1/Y_2$ domains. In contrast, in $H_{c2}$ with lower criteria, the additional peaks are absent because the sample resistivity near the zero resistance state is mostly governed by the domain with the highest volume fraction. Nevertheless, even for $H_{c2}$ with the lower criteria (e.g. $R/R_n = 5\%$), there is noticeable strain dependence near $\phi_{ab} = 0°$. This dependence is also attributable to the domain change by comparison with a model simulation explained later.

Here we briefly discuss another possible interpretation for the change in the $H_{c2}$ anisotropy: a strain-induced crossover from an un-pinned single nematic domain state to a strongly-pinned single domain state. Theoretically, sixfold sinusoidal $H_{c2}$ has been proposed for a single-domain state in the ideal situation of a complete absence of nematicity pinning, and twofold $H_{c2}$ for finite nematicity pinning[34], which might have been achieved by the application of uniaxial strain. However, this scenario is less likely compared to the multi-domain scenario, because of the following several reasons. Firstly, if the sixfold $H_{c2}$ originates from a single domain, it is difficult to explain the fact that the sixfold behavior is accompanied by a change in the broadness of the superconducting transition. The superconducting transition in magnetic field is rather broad for $\phi_{ab} = \pm 90°$ without the uniaxial strain, whereas the transition becomes sharper under

compressive strain (see Fig. 1c); and the sixfold behavior was observed only near the onset of the transition. This is difficult to be explained by a single domain scenario, since the transition width would be insensitive to the field direction or to the applied strain. Secondly, the observed onset $H_{c2}$ does not exhibit a perfect sixfold behavior: the onset $H_{c2}$ vs $\phi_{ab}$ curve exhibits stronger peak at $\phi_{ab} = \pm 90°$ than the peaks at $\pm 30°$ or $\pm 150°$. In contrast, for an un-pinned domain, we expect that $H_{c2}$ should be perfectly sixfold symmetric[34]. Indeed, we demonstrate that the observed $H_{c2}$ cannot be fitted with a simple sixfold sinusoidal function even for the onset $H_{c2}$ with strongest sixfold component, as shown in Supplementary Fig. 15. This imperfectness of the sixfold behavior in the onset $H_{c2}$ is better explained by the multi-domain scenario. Thirdly, we find that, with increasing compressive strain, there is a smooth and anisotropic disappearance of the sixfold pattern. In particular, the pair of satellite peaks at $-150°$ & $+30°$ and $-30°$ & $+150°$ have different strain dependences; as shown in Supplementary Fig. 16, the former pair is still visible even under $-1.19\%$ strain whereas the latter pair completely disappears. If we were to assume that the sixfold component of $H_{c2}$ is due to a single sixfold domain, then with the application of strain $H_{c2}$ would suddenly exhibit a twofold behavior and both satellite peaks should disappear simultaneously. Nevertheless, we cannot completely

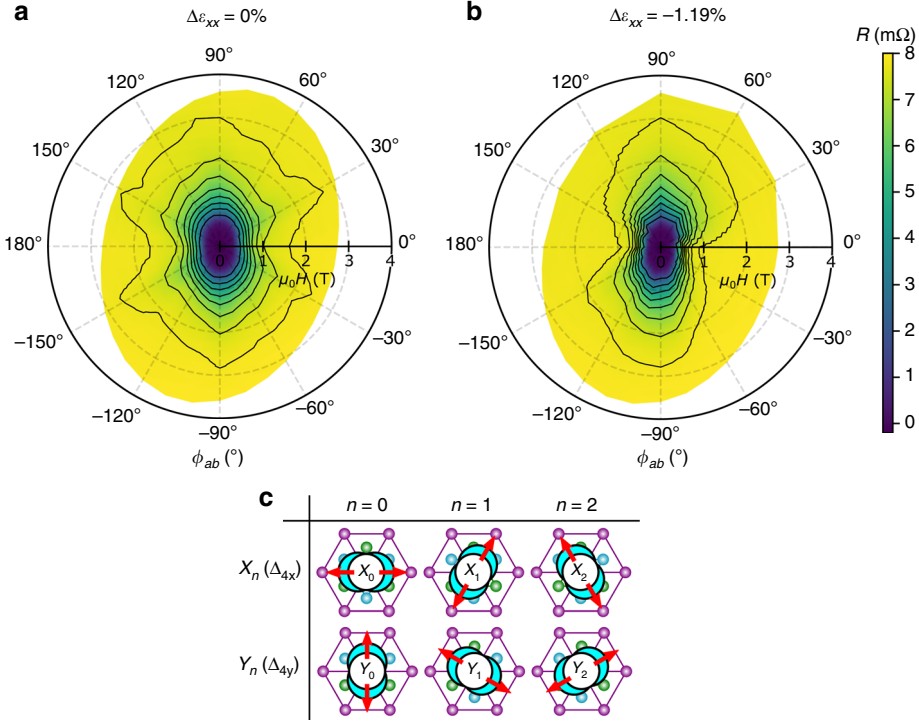

**Fig. 2 Disappearance of nematic superconducting domains in Sr$_x$Bi$_2$Se$_3$ under compressive strain. a b** Color polar plot of magnetoresistance for $H\|ab$ measured at the relative strains of $\Delta\varepsilon_{xx} = 0$ (**a**) and $-1.19\%$ (**b**) with 250 $\mu$A applied current and 2.2 K. The light-green regions extending along $\pm30$ and $\pm150°$ in **a** indicate existence of nematic subdomains, which substantially disappears under applied strain (**b**). The contours are drawn from 0.5 m$\Omega$ to 7.5 m$\Omega$ in steps of 1 m$\Omega$. **c**, Table of the 6 possible nematic superconducting states that can exist in the sample as domains. $X_n$ and $Y_n$ ($n = 0, 1, 2$) domains exhibit $\Delta_{4x}$ and $\Delta_{4y}$ states with the large $H_{c2}$ along one of the $a$ axes ($\phi_{ab} = (60n)°$) and $a^\star$ axes ($\phi_{ab} = (90 + 60n)°$), respectively, as indicated with the red arrows. The crystal structure in the $ab$ plane of Bi$_2$Se$_3$ is shown with the schematic superconducting wave function in its center. The thickness of the blue crescent depicts the superconducting gap amplitude. Source data are provided as a Source Data file.

deny a possibility that some part of the minor domains exhibits the sixfold $H_{c2}$ behavior expected for the ideal situation, as a result of accidental cancellation of pre-existing symmetry breaking field and externally applied strain. But this contribution should be, if exists, small and is neglected in the analyses in the following. Lastly, for a realistic situation with certain non-uniformity in the sample, $H_{c2}$ defined using higher $R/R_n$ values pick up the most robust superconducting regions and the effective uniformity is enhanced.

**Model simulation of the upper critical field.** In order to confirm our scenario, we performed a simple model simulation for $H_{c2}$. In this simulation, we assume a network consisting of many $Y_0$ domains and one of each $Y_1/Y_2$ domain and calculate the net resistance under magnetic fields ("Methods" section). Then $H_{c2}$ with various criteria is evaluated from the calculated resistivity curves. Under strain, the minor domains are assumed to change into $Y_0$ domains. We should comment here that our model is free from the choice of the detailed mechanism of appearance of finite resistivity under magnetic fields near $H_{c2}$. This is because we model the behavior of each domain by making use of experimental magnetoresistance data under high strain, and also because we used classical equations for circuit resistance calculation (Supplementary Note 10).

As shown in Fig. 4, the simulation reproduces all the observed features described above, even without any change of $H_{c2}$ and in-plane $H_{c2}$ anisotropy in each domain, although the model is quite simple. We have tried several other circuit configurations and confirmed that the result of the simulations are rather similar, as shown in Supplementary Fig. 17. We find that setting slightly smaller $H_{c2}$ (ca. decrease of 10%; the broken curves in Fig. 4a) of

the $Y_0$ domain under strain gives a better match with the experimental data. Separately, we also performed fitting of this model to the experimental onset $H_{c2}$. As shown in Supplementary Fig. 15, the fitting was reasonably successful. These findings lead one to infer that the circuit model is reasonable and the observed behavior is almost solely explained by the change of the nematic SC subdomains.

**Summary of results.** Summarizing our findings, we succeeded in repeatable in-situ uniaxial-strain control of nematic SC domains in Sr$_x$Bi$_2$Se$_3$, covering pre-existing tensile regime to the compressive regime. The primary effect of the increasing compressive strain is the suppression of minor $Y_1/Y_2$ domains, well reproduced by a simple model simulation. Other properties are rather insensitive to the strain, but there are decreasing trends in $T_c$ as well as $H_{c2}$ of the main domain under compression. We comment here that, in most superconductors, domains with different but degenerate SC order parameters are absent. The only exception has been chiral SC domains with opposite Cooper-pair angular momenta suggested for the chiral-superconductor candidate Sr$_2$RuO$_4$ but their existence and control are not yet confirmed[35]. The present study provides the first demonstration of SC-domain engineering, a superconductor counterpart of domain engineering common in magnets.

## Discussion
The coupling between the nematic superconductivity and uniaxial strain has been proposed using the Ginzburg–Landau (GL) formalism[34,36]. The strain couples to the nematic superconductivity through the free energy $F_\varepsilon = g[(\varepsilon_{xx} - \varepsilon_{yy})(|\eta_x|^2 - |\eta_y|^2) + 2\varepsilon_{xy}$

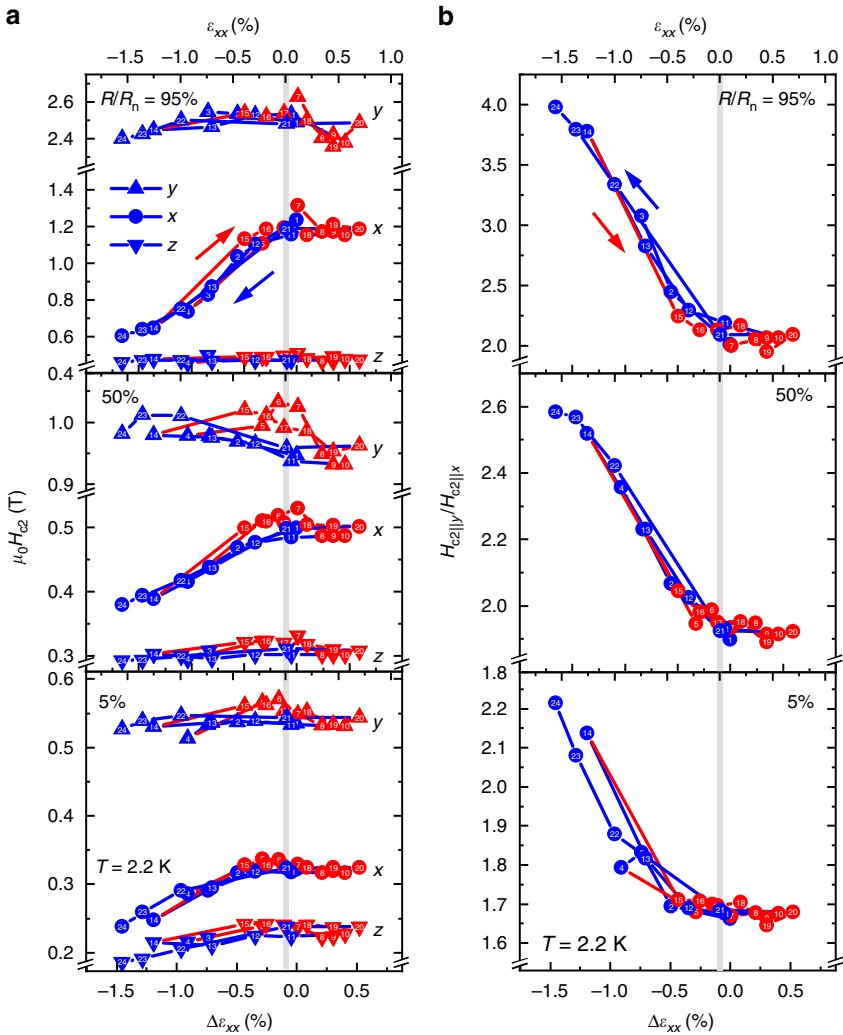

**Fig. 3 Reversible uniaxial-strain control of the nematic superconductivity. a** Upper critical field ($H_{c2}$) at 2.2 K along the $a$ axis ($x$; $\phi_{ab} = 0°$; circles), $a^*$ axis ($y$; $\phi_{ab} = -90°$; upper triangles), $c$ axis ($z$; lower triangles), as a function of the relative strain $\Delta\varepsilon_{xx}$ induced by an applied voltage to the piezostacks. **b**, In-plane $H_{c2}$ anisotropy $H_{c2\|y}/H_{c2\|x}$ as a function of strain. At the top of each panel, the estimated actual strain $\varepsilon_{xx} \simeq \Delta\varepsilon_{xx} + 0.1\%$ is indicated (see "Methods" section) and the gray region illustrates the possible range in the actual zero strain. The numbers in the top corner of each sub-panel indicates the criteria used for determining $H_{c2}$ (see "Methods" section). The numbers in the data points indicate the order of the measurements. The blue and red data points indicate the cases that the measurement was performed after a decrease and increase in strain, respectively. Here, the $H_{c2}$ anisotropy varies systematically with external strain. Source data are provided as a Source Data file.

$(\eta_x\eta_y^* + \eta_y\eta_x^*)]$, where $g$ is the coupling constant, $\eta_x$ and $\eta_y$ are the amplitudes of the $\Delta_{4x}$ and $\Delta_{4y}$ components. Here we considered only the lowest-order coupling terms for simplicity. However, this treatment should be valid when considering the situation near $T_c$. This relation indicates that uniaxial $\varepsilon_{xx}$ strain prefers one of the $\Delta_{4x}$ or $\Delta_{4y}$ states, depending on the sign of $g$. If a pre-existing symmetry breaking field exists, the nematic SC order parameter is initially fixed to the pre-existing field direction but eventually the state most favored by the external strain direction will be chosen with increasing strain. This is true even when the pre-existing field and external strain have a finite angle, as in the case for the $Y_1$ or $Y_2$ domains: the nematicity gradually rotates toward the external strain (see Supplementary Fig. 8). However, in these phenomenological theories, the sign of $g$ remains arbitrary and should be determined based on experiments. Our result, a multi-domain sample driven to a mono-domain $\Delta_{4y}$ state by $\varepsilon_{xx} < 0$ as shown in Fig. 3c, indicates that $g$ is negative, an important step toward modeling of the nematic SC phenomenon. Moreover, this negative $g$ provides a crucial constraint to realistic microscopic theories on the pairing

mechanism. Such model should also explain the observed weak sensitivity of $T_c$ on $\varepsilon_{xx}$. For example, a proposed odd-parity fluctuation model making use of phonons dispersing along the $k_z$ direction[37–40] can be compatible with our observation, since such $k_z$ phonons should be less sensitive to the in-plane distortions.

Our results also provide information about the magnitude of $g$. The present work shows that compressive strain of around $-1\%$ is required to alter the nematic SC domains. This is unexpectedly much larger than the strain required to homogenize the normal-state nematicity in iron pnictides (typically around 0.01–0.1%[3]). Thus the nematicity-lattice coupling is much weaker in doped $Bi_2Se_3$ than in iron-based superconductors. This difference may be due to the difference in the size of the nematic ordering elements. In nematic superconductors, the nematicity is carried by Cooper pairs, which are non-local objects with sizes much larger than inter-atomic distances. In contrast, in iron pnictides, the nematicity occurs due to orbital ordering, which occurs within one atomic site. We speculate that large strain is required to control a large and non-local object.

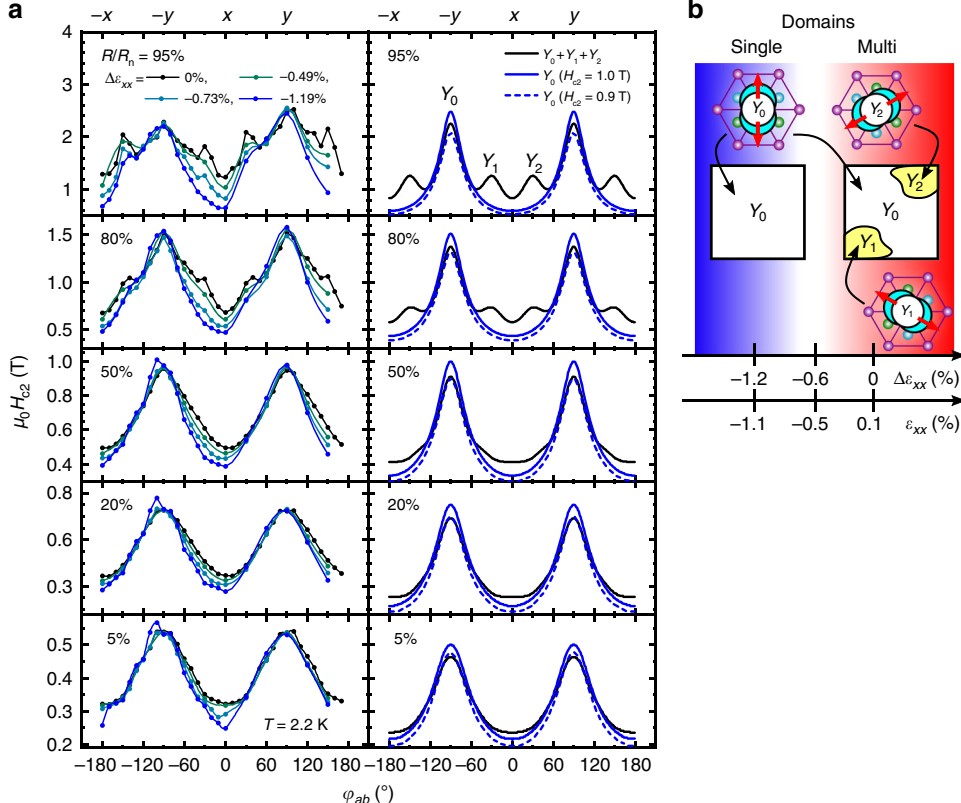

**Fig. 4 Evidence for the control of nematic superconducting domains with strain. a** In-plane field angle $\phi_{ab}$ dependence of the upper critical field $H_{c2}$ at 2.2 K for various criteria (see "Methods" section), which are indicated with a number in the top-left corner of each sub-panel. The curves colored from black to blue are in the order of increasing compressive strain; the numbers in the legend indicate the value of $\Delta\varepsilon_{xx}$. The results of a simulation are shown in the second column (see "Methods" section) capturing key features of the observation. **b** Schematic showing spatial configurations of nematic superconducting domains controlled in-situ by the uniaxial strain in our experiment. The yellow regions are the minor domains ($Y_1$ and $Y_2$), which are suppressed by the application of compressive strain, as evidenced by the changes in the $H_{c2}(\phi_{ab})$ curves. Source data are provided as a Source Data file.

Coming back to the GL theories, they predict that $T_c$ linearly increases with increasing strain in either tensile or compressive directions, accompanied by a kink in $T_c(\varepsilon_{xx})$ at the strain where the nematic state changes between $\Delta_{4x}$ and $\Delta_{4y}$[34,36]. This prediction, at first glance, seems to be inconsistent with our decreasing trend of $T_c$ with increasing $|\varepsilon_{xx}|$. However, we should note that $T_c$ of doped Bi$_2$Se$_3$ decreases under hydrostatic pressure, i.e. under isotropic strain[10]. This effect is not taken into account in the above mentioned GL free energy, which couples only to the anisotropic strains. In the actual experiments, a combination of the increasing and decreasing trends in $T_c$ due to anisotropic and isotropic strains, respectively, is observed. If the latter is relatively stronger, the observed small decrease of $T_c$ by compressive strain is explained. Moreover, the existence of multiple domains weakens the predicted kink in $T_c(\varepsilon_{xx})$, because each domain's $T_c(\varepsilon_{xx})$ curve convolves. This will result in a rounded kink, further obfuscating the linear behavior predicted from a mono-domain model.

Before concluding, we compare our findings with recent and related works on doped Bi$_2$Se$_3$ superconductors. Kuntsevich et al. found that their Sr$_x$Bi$_2$Se$_3$ samples grown using the Bridgman method exhibits small lattice distortions (0.02% in-plane elongation and 0.005° $c$-axis inclination) at room temperature and weak twofold angular magnetoresistance (1–4%) even in the normal state[16]. In contrast, Sr$_x$Bi$_2$Se$_3$ samples grown using melt-growth technique, as the samples we used in this work, does not seem to have detectable lattice distortions or normal-state nematicity in resistivity[11,13] (Also see Supplementary Note 13). Regarding the normal-state nematicity, we should mention here

that twofold behavior in the magnetic-field-angle dependence of specific heat of Sr$_x$Bi$_2$Se$_3$ has been recently reported using samples grown with a self-flux method (equivalent to the melt growth method)[17]. Nevertheless, the origin of large normal-state specific-heat oscillation requires further consideration.

More recently, Kuntsevich's group examined the dependence of superconducting nematicity on the as-grown lattice distortion using their Bridgman-grown samples[18]. They find that $H_{c2}$ is larger along the $x$ axis for a sample with $x$-axis compressive strain, and it is larger along $y$ under $x$-axis elongation. This tendency is, interestingly, opposite to the findings of our present work. Given that the strain range investigated in their work (±0.04%) is much smaller than that in our work (+0.1% to −1.2%), the origin of the nematicity in the samples with as-grown strain may be attributed to another underlying symmetry breaking field (e.g. alignment of intercalated Sr), rather than a simple uniaxial lattice distortion.

In Nb$_x$Bi$_2$Se$_3$, a spontaneous lattice distortion occurring at a temperature slightly above $T_c$ has been experimentally reported[41], apparently consistent with the theoretical proposal of vestigial nematic order induced by superconducting fluctuation[42]. In this experiment, $H_{c2}$ along $x$ is found to become larger (corresponding to the $\Delta_{4x}$ state) after the lattice spontaneously shrinks along the $y$ ($a^*$) axis, inferring a negative coupling constant $g$. This observation, obtained with passive lattice response, is consistent with and complementary to the observation of the present work using active lattice control.

To conclude, we provide the first experimental demonstration of uniaxial-strain control of nematic superconductivity in doped Bi$_2$Se$_3$. Firstly, the $x$-axis compression suppresses minor domains

while stabilizing the $\Delta_{4y}$ state. Secondly, we determined the sign of the nematic coupling constant. These findings should provide bases toward resolving open issues of this highly attractive superconductor. Additionally, this work points to possible engineering of topological nematic superconductivity by uniaxial strain.

## Methods

**Sample preparation and characterization.** Single crystals of $Sr_xBi_2Se_3$ (nominal $x = 0.06$) were grown from high-purity elemental is Sr chunk (99.99%), Bi shot (99.9999%), and Se shots (99.9999%) by a conventional melt-growth method. The raw materials were mixed with a total weight of 5.0 g and sealed in an evacuated quartz tube. The tube was heated to 1223 K and kept for 48 h with intermittent shaking to ensure the homogeneity of the melt. Then it was cooled slowly to 873 K at a rate of 4 K h$^{-1}$ and finally quenched into ice water. It is worth pointing out that quench is essential for obtaining superconducting samples with high shielding fraction. The sample used here was cut from a large shiny crystal by wire saw, and the size is 4 mm (length) × 0.53 mm (width) × 0.5 mm (thickness; along the $c$ axis) with the longest dimension along the $a$ axis.

**Strain cell and sample mounting.** We constructed a custom-made piezoelectric-based uniaxial strain cell (ref. [31]), based on the design of ref. [32]. The bar-shaped sample was mounted between two anvils by a strong epoxy (Stycast 2850FTJ, Henkel Ablestik Japan Ltd.). The anvils can apply compressive or tensile strain on the sample by applying a positive voltage on the inner or outer piezo stacks, respectively. Thus the strain was applied parallel to the $a$ axis, as shown in Fig. 1. The maximum applied voltage range for each piezo stack was $-400$–$600$ V corresponding roughly to $-13\,\mu m$ to $20\,\mu m$ length changes of the piezo stacks used (P-885.11, PI) at cryogenic temperatures. A parallel-plate capacitor was mounted on the anvils to track the distance between the two plates by measuring the capacitance using a capacitance bridge (2500A, Andeen-Hagerling). The strain was then determined by the displacement divided by the exposed sample length, which was $1.14 \pm 0.05$ mm in this study.

**Estimation of the thermally-induced strain.** The effect of thermal contraction of the sample and the strain cell should be taken into consideration. Because the materials used in the strain cell are placed symmetrically between the compressive and tensile arms, the thermal strain on the sample originates only from the asymmetric part[31]; on the compressive arm, the sample with the length $L_{sample}$ of 1.14 mm is placed, but on the tensile arms there are Ti blocks. This 1.14-mm length Ti part shrinks less than the sample, resulting in a tensile strain to the sample after cooling down from the epoxy curing temperature (around 350 K). The shrinkage of the sample $\Delta L_{sample}/L_{sample}$ is evaluated as $[a(4\,K) - a(350\,K)]/a(350\,K) = -0.36\%$. Here, we used the lattice constants of $Bi_2Se_3$ reported in ref. [43]. We note that $a(4\,K)$ and $a(350\,K)$ is estimated by a linear extrapolation because ref. [43] reports $a$ values only between 10 K and 270 K. For Ti, the shrinkage $\Delta L_{Ti}/L_{Ti}$ is evaluated to be $-0.23\%$ by integrating the linear thermal expansion coefficient between 4 K and 350 K reported in ref. [44]. The thermal expansion coefficient at 4 K and 350 K were obtained after linearly extrapolated. Thus, the thermally-induced strain to the sample is tensile and $(\Delta L_{Ti} - \Delta L_{sample})/L_{sample} = +0.13\%$ considering $L_{Ti} = L_{Sample}$. In addition, because of the stiffness of the component materials, in particular the epoxy, the actual strain transmitted to the sample may be reduced by roughly 56%[31]. Thus, the value $+0.13\%$ should be considered as the upper bound, and the lower bound should be $0.13\% \times 0.56 = 0.07\%$. To conclude, by taking the average, the thermally-induced strain is evaluated to be $0.10 \pm 0.03\%$: the actual strain $\varepsilon_{xx}$ is given as $\varepsilon_{xx} \simeq \Delta\varepsilon_{xx} + 0.1\%$, where $\Delta\varepsilon_{xx}$ is the strain applied relative to the situation of zero applied voltage to the piezo stacks.

**Resistivity measurement.** Sample resistivity was measured by four-terminal sensing: we applied a DC current using a current source (6221, Keithley Instruments) to the two outer wires and measure the resultant voltage by a nanovoltmeter (2182A, Keithley Instruments) on the inner two wires. To subtract the voltage offset, we use the Delta Mode of the combined operation of these instruments: the polarity of the current was periodically changed to acquire only the voltage component that is dependent on the current. We mostly used 250-$\mu A$ current for resistivity measurements. We have checked that Joule heating or any other current-induced suppression of superconductivity is absent with 250 $\mu A$, by comparing resistivity behavior under various current (Supplementary Note 12). Au wire (20 $\mu m$ diameter) were directly connected to the $ac$ surface of the sample by Ag paint (4929N, Du Pont). To improve the mechanical stability of the sample, the Au wires were anchored onto the $ab$ surface by Ag epoxy (H20E, EPOTEK), which has been confirmed to be electrically insulating to the sample. The four contacts were equispaced by about 0.2 mm. The contact resistance was on the order of 100 $\Omega$ at room temperature.

**Temperature and magnetic-field control.** We used a $^3He/^4He$ dilution refrigerator (Kelvinox 25, Oxford Instruments) to cool down the sample. It was inserted into the vector magnet described below. The lowest temperature achievable is roughly 80 mK, well below the superconducting transition temperature of ~2.8 K. The temperature was measured using a resistive thermometer (Cernox, Lakeshore) and a resistance bridge (AVS-47, Picowatt). A 350-$\Omega$ strain gauge (KFG-1-350-C1-16, KYOWA), that was used as a heater for temperature control, was mounted close to the strain cell.

We applied the magnetic field using a vector-magnet system[45], which consists of two orthogonal superconducting magnets (pointing in the vertical and horizontal directions in the laboratory frame) inside a dewar that rests on a horizontal rotation stage. This system allows us to direct the magnetic field accurately in any direction in space while the refrigerator, as well as the sample, is fixed. The superconducting magnets can apply fields up to 3 T (vertical) and 5 T (horizontal). The magnetic field can be controlled with a resolution of 0.1 mT. The precision of the horizontal rotation of the helium dewar is 0.001°, with negligible backlash. The strain cell was fixed with a sample mounted, so that the $a$ axis is roughly along the vertical direction in the laboratory frame. The precise directions of the crystalline axes with respect to the laboratory frame are determined by making use of the anisotropy in $H_{c2}$. Once the directions of the crystalline axes are determined, we can rotate the magnetic field within the sample frame. All field angle values presented in this Article are defined in the sample frame. Refer to Supplementary Note 8 and Supplementary Fig. 9 for the detailed mathematical explanation for the vector transformations and a demonstration of the field alignment. In addition, see Supplementary Note 9 for the rationale behind the choice of temperature and magnetic field value for the alignment.

**Evaluation of the upper critical field.** The upper critical field $H_{c2}$ was evaluated by the value of the magnetic field at which the sample resistivity reaches a certain percentage of the normal-state resistivity. If the resistivity value falls in between two data points, then $H_{c2}$ is determined by using linear interpolation. For a more detailed methodology see Supplementary Note 4. Temperature dependence of $H_{c2}$ and its anisotropy is given in Supplementary Note 11.

**Model simulation.** The experimental $H_{c2}(\phi_{ab})$ curve shows three peaks indicating three nematic domains. Thus, we simulated the $H_{c2}$ behavior of multi and single-domain samples by considering an electrical circuit consisting of a network of resistive elements representing the three possible nematic SC domains. For the simulation shown in the main text, the circuit is assumed to be a 3D network (see Supplementary Fig. 13) of 12 elements to model the situation that current passes from end to end through a 3D distribution of domains. For the multi-domain simulation corresponding to the non-strained sample, the 12 elements are divided into 10 $Y_0$ nematic domains, and one of each $Y_1$ and $Y_2$ domains. The exact positions are described in Supplementary Note 10. For the single-domain simulation corresponding to the highly compressed sample, all the 12 elements are assumed to be $Y_0$ domains. The calculation of $H_{c2}$ is done as follows: firstly, for a fixed $H$ and $\phi_{ab}$, resistivity of each circuit element is calculated from an empirical relationship among resistance, applied magnetic field $H$ and field direction $\phi_{ab}$, by taking into account $H_{c2}$ anisotropy of each domain (see Supplementary Note 10 for details). Secondly, the total circuit resistance of the network $R_{total}$ is calculated. The first and second step is iterated while varying $H$ and $\phi_{ab}$, to obtain the $H$ dependence of $R_{total}$ for each $\phi_{ab}$. Lastly, $H_{c2}$ at $\phi_{ab}$ is determined from the $R_{total}(H)$ curve at $\phi_{ab}$ using the same method as that used for the experimental data analysis (see Supplementary Note 4).

## Data availability

The source data underlying Figs. 1c, 2a-c, 3a-b, 4a, and Supplementary Figs. 1–7, 9, 10, 12, 18–20 are provided as a Source Data file. The other data that support the findings of this study are available from the corresponding author upon request.

## Code availability

The codes used for the simulation are available from the corresponding author upon request.

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

## Acknowledgements

The authors acknowledge H.-H. Wen, J. Schmalian, P. T. How, S.-K. Yip, V. Kozii, and J. W. F. Venderbos, H.-S. Xu, and G. Mattoni for valuable discussions. We also acknowledge C. W. Hicks, M. E. Barber, A. Steppke, F. Jerzembeck, and A. P. Mackenzie for sharing their knowledge into the construction of a strain cell. This work was supported by JSPS Grant-in-Aids for Scientific Research on Topological Materials Science (KAKENHI JP15H05851, JP15H05852, JP15H05853, JP15K21717), on Quantum Liquid Crystals (KAKENHI 20H05158), by JSPS Grant-in-Aid KAKENHI 17H04848, and by the JSPS Core-to-Core Program. The work at Kyoto was also supported by ISHIZUE 2020 of Kyoto University Research Development Program. The work at Cologne was funded by the Deutsche Forschungsgemeinschaft (DFG, German Research Foundation) - Project number 277146847—CRC 1238 (Subproject A04).

## Author contributions

This study was designed by I.K., S.Y., and Y.M.; I.K. performed resistivity measurements and analyses, with the assistance of S.Y. and the guidance of Y.M.; Z.W. and Y.A. grew single crystalline samples and characterized them. The uniaxial strain cell was designed and constructed by I.K.; The manuscript was prepared mainly by I.K. and S.Y., based on discussion among all authors.

## Competing Interests

The authors declare no competing interests.
