## [Peer Review File · Nature Communications]

Reviewers' comments:

Reviewer #1 (Remarks to the Author):

Authors study Hc2 in doped BiSe, a putative odd parity superconductor with multi-component order parameter. There are very few superconductors of such kind; in addition to being fundamentally interesting, they may also be useful in the context of quantum information systems, since they can host Majorana fermions.

Earlier experiments have shown rotational symmetry breaking in Hc2, down to two fold, which was taken as an indication of nematic superconductivity (since there is no visible structural or resistive anisotropy in the normal state). Unfortunately, earlier experiments failed to see reorientation of the anisotropy either by thermal cycling or strain. In the present experiment, authors give compelling evidence that this may be possible.

The experiment shows that in the absence of externally applied strain (there is probably build-in strain, however), there is a signature of six-fold symmetry in the Hc2 pattern. In theory, Hc2 is only sensitive to the second order terms in Ginzburg-Landau functional. For one component superconductors in tetragonal, trigonal, or hexagonal systems there can therefore be no anisotropy in Hc2. The only way anisotropy can appear is if the crystal itself has only two-fold symmetry, or if the order parameter has multiple components. In the former case, Hc2 could have 2 fold anisotropy, but there is no reason to have any sign of six fold, as seen in the present experiment. This observation thus provides a compelling support for two-component odd-parity superconductivity.

While I find the experimental results compelling, I do not agree with the interpretation in terms of Y domains. As stated above (see <https://arxiv.org/pdf/1603.03406.pdf>), the six-fold Hc2 anisotropy *is not* due to domain formation, since it is a result of *linear* analysis of GL equations. Indeed, experimentally, the six-star pattern only appear at large field in Fig 2a, and at low fields (blue center) the pattern is only 2 fold -- indicating a single domain!

Application of strain pushes the system apparently in the same direction as the built-in strain, making the two-fold anisotropy at small fields more pronounced, while removing the six-fold terms at high fields. This is consistent with an evolution between Fig 1 and 2 in <https://arxiv.org/pdf/1603.03406.pdf>.

To summarize, I find the experiment excellent, worth publishing in Nature Communications; however, the modeling and interpretation have to be improved.

Ivar Martin

Reviewer #2 (Remarks to the Author):

The manuscript by Kostylev et al. provides a detailed study on the effect of uniaxial strain on the magnetoresistance anisotropy in the mixed states of Sr-doped Bi2Se3. The approach is interesting. I am generally confident with their experimental approach and admire the detail they pursued the project. However, I have a few questions about the analysis.

How does the normal state resistivity change under strain? The determination of the Hc2 depends on the normal state resistivity. Fig. 3 also depends on the resistivity normalized by the normal state resistivity.

There is always a question about if the nematic order originates from the superconducting state or the electronic state anisotropy in the normal state. I do not quite get how the manuscript excludes this possibility.

I am puzzled too by the model based on the G-L approach and multiple domains. The change of resistivity in the fluctuation regime is sensitive to the parameters. I am concerned that the model is too much overanalysis.

Therefore, I would suggest a revision of the manuscript before the consideration for acceptance of the manuscript.

Reviewer #3 (Remarks to the Author):

In the paper by Kostylev et al. a detailed study of the strain dependence of the nematic superconductivity in $\text{Sr}_x\text{Bi}_2\text{Se}_3$ is presented. The nematic superconductivity is monitored by the anisotropy in the upper critical field. This is a timely topic in the center of the field of topological superconductors and certainly of sufficiently broad scientific interest for Nature Communications. The paper is well written and the data and analysis are sound. The authors present the first control of superconductivity via uniaxial strain in doped Bi_2Se_3 superconductors, in that sense it is pioneering work. The experiments are properly described and the outcome is clear. The angular variation of the resistance is compared with a simulation to provide a further understanding of the data.

The quality of the paper is high and I expect it to have a significant impact. I recommend publication in Nature Communications. Below are a few minor comments that should be addressed prior to publication.

1. On page 1 line 6 "are" should read "is" and on page 2 line 25 "have" should read "has".
2. On page 3, line 42 the sample to sample variation is discussed. Here a reference to the paper by Kuntsevich in New Journal of Physics (20-2018-103022) is missing.
3. Kuntsevich et al. also published strain dependence in $\text{Sr}_x\text{Bi}_2\text{Se}_3$ probed by the combination of XRD and transport. the authors should make a link and comparison to this work (Phys. Rev. B 100-20190224509).
4. I am puzzled by the change of T_c in Fig.S3 when the current is decreased from 250 to 50 μA . Normally such differences can be attributed to Joule heating of the sample for the larger current (the sample then has a higher temperature than the thermometer). The authors should explain whether local heating can be excluded. Overall the current in all experiments is large (250 μA) and absence of Joule heating should be demonstrated.

Response to Reviewers' comments

Reviewer #1

[Comment 1 of Reviewer #1]

Authors study H_{c2} in doped BiSe, a putative odd parity superconductor with multi-component order parameter. There are very few superconductors of such kind; in addition to being fundamentally interesting, they may also be useful in the context of quantum information systems, since they can host Majorana fermions.

Earlier experiments have shown rotational symmetry breaking in H_{c2} , down to two fold, which was taken as an indication of nematic superconductivity (since there is no visible structural or resistive anisotropy in the normal state). Unfortunately, earlier experiments failed to see reorientation of the anisotropy either by thermal cycling or strain. In the present experiment, authors give compelling evidence that this may be possible.

[Our response]

We thank Prof. Ivar Martin, Reviewer #1, for taking time to carefully review our manuscript and for providing valuable comments. Our responses to the comments and corresponding revisions are described below. For your convenience, we submit another manuscript file, in which revisions are indicated in red.

[Comment 2 of Reviewer #1]

The experiment shows that in the absence of externally applied strain (there is probably build-in strain, however), there is a signature of six-fold symmetry in the H_{c2} pattern. In theory, H_{c2} is only sensitive to the second order terms in Ginzburg-Landau functional. For one component superconductors in tetragonal, trigonal, or hexagonal systems there can therefore be no anisotropy in H_{c2} . The only way anisotropy can appear is if the crystal itself has only two-fold symmetry, or if the order parameter has multiple components. In the former case, H_{c2} could have 2 fold anisotropy, but there is no reason to have any sign of six fold, as seen in the present experiment. This observation thus provides a compelling support for two-component odd-parity superconductivity.

*While, I do not agree with the interpretation in terms of Y domains. As stated above (see <https://arxiv.org/pdf/1603.03406.pdf>), the six-fold H_{c2} anisotropy *is not* due to domain formation, since it is a result of *linear* analysis of GL equations. Indeed, experimentally, the six-star pattern only appear at large field in Fig 2a, and at low fields (blue center) the pattern is only 2 fold -- indicating a single domain!*

Application of strain pushes the system apparently in the same direction as the built-in strain, making the two-fold anisotropy at small fields more pronounced, while removing the six-fold terms at high fields. This is consistent with an evolution between Fig 1 and 2 in <https://arxiv.org/pdf/1603.03406.pdf>.

[Our response]

We first appreciate Prof. Martin for providing a very positive comment "*I find the experimental results compelling*".

We thank Prof. Martin for pointing out another possible interpretation, namely a uniaxial-strain-induced crossover from a weakly-pinned single nematic domain state to a strongly-pinned single domain state. Indeed, theoretically, six-fold H_{c2} has been proposed for a single-domain state with weak nematicity pinning. However, we discuss below that this scenario is less likely compared with the multi-domain scenario we propose, because of the following two reasons:

Firstly, if the six-fold H_{c2} originates from a single domain, then it is difficult to explain the fact that the six-fold behavior is accompanied by a change in the broadness of the superconducting transition. The superconducting transition in magnetic field is rather broad for $\varphi = \pm 90^\circ$ without the uniaxial strain, whereas the transition becomes sharper under compressive strain (See Fig. 1c); and the six-fold behavior was observed only near the onset of the transition. For a single domain scenario, we expect that the transition width would be insensitive to

the field direction or to the applied strain. In order to explain the six-fold behavior near the onset, we need to consider a somewhat special situation where the order parameter starts to be pinned between the onset H_{c2} and the zero-resistance H_{c2} . The multi-domain scenario explains the broad transition more naturally; resistivity just below the onset H_{c2} does not reach zero because the superconducting percolation path is not formed between the voltage leads due to the multi-domain nature.

Secondly, the observed onset H_{c2} does not exhibit a perfect six-fold behavior: the onset H_{c2} vs φ curve exhibits stronger peaks at $\varphi = \pm 90^\circ$ than the peaks at $\pm 30^\circ$ or $\pm 150^\circ$. In contrast, for a weakly-pinned single-domain scenario, we expect that H_{c2} should be perfectly six-fold symmetric. This “imperfectness” of the six-fold behavior in the onset H_{c2} is better explained by the multi-domain scenario.

Nevertheless, this point raised by Prof. Martin is quite important. Thus, to address this issue, we added a paragraph in pages 6-7 of the main text describing and discussing this weakly-pinned single-domain scenario.

[Comment 3 of Reviewer #1]

To summarize, I find the experiment excellent, worth publishing in Nature Communications; however, the modeling and interpretation have to be improved.

Ivar Martin

[Our response]

We again thank Prof. Martin for the very positive evaluation and feedback. With the response and revision described above, we now believe now this manuscript is acceptable for publication.

Reviewer #2

[Comment 1 of Reviewer #2]

The manuscript by Kostylev et al. provides a detailed study on the effect of uniaxial strain on the magnetoresistance anisotropy in the mixed states of Sr-doped Bi₂Se₃. The approach is interesting. I am generally confident with their experimental approach and admire the detail they pursued the project. However, I have a few questions about the analysis.

[Our response]

We thank Reviewer #2 for careful reviewing our manuscript. We are pleased that they evaluated our work quite positively as “*The approach is interesting. I am generally confident with their experimental approach and admire the detail they pursued the project*”. Our responses to the comments and corresponding revisions are described below. For your convenience, we submit another manuscript file, in which revisions are indicated in red.

[Comment 2 of Reviewer #2]

How does the normal state resistivity change under strain? The determination of the H_{c2} depends on the normal state resistivity. Fig. 3 also depends on the resistivity normalized by the normal state resistivity. There is always a question about if the nematic order originates from the superconducting state or the electronic state anisotropy in the normal state. I do not quite get how the manuscript excludes this possibility.

[Our response]

We thank Reviewer #2 for pointing out this issue. Actually, the change in the normal-state resistivity by the uniaxial strain was negligible. To describe this, we prepared a new figure as shown below (Fig. R1; also added to the Supplementary Information as Fig. S17), comparing field-angle dependence of the normal state resistivity with and without the uniaxial strain.

Firstly, one can see that the two-fold behavior characterizing normal-state nematicity is absent, although there is a weak one-fold (i.e. 360-deg-periodic) behavior of unknown origin. Thus, the observed large two-fold behavior should in the superconducting state originate from the emergence of nematic superconductivity. Secondly, it is clear that the uniaxial-strain effects to the normal-state resistivity is practically absent (except for a small constant increase). Thus the observed strong change of resistivity in the superconducting regime by strain application should be solely due to the control of nematic superconductivity.

Figure R1: Comparison of the angular magnetoresistance in the normal state measured with and without strain. Although a weak one-fold change, whose origin is not known, was observed, two-fold component is almost absent in the data measured without strain (black data points). This fact indicates that the normal-state nematicity is rather weak for this sample. Moreover, there is no change in the angular dependence even after application of compressive strain (blue data points).

Since this point is quite important, we added a sentence in the Main Text (page 4) and a new subsection in the Supplementary Information (Supplementary Note S13) with a new figure Fig. S17.

[Comment 3 of Reviewer #2]

I am puzzled too by the model based on the G-L approach and multiple domains. The change of resistivity in the fluctuation regime is sensitive to the parameters. I am concerned that the model is too much overanalysis.

[Our response]

First of all, we agree that the resistivity near the superconducting transition is quite sensitive to various parameters and that one should carefully avoid overanalyses. On the other hand, we believe that it is quite important and valuable to provide a simple model that can grasp the essence of the observed behavior.

We here presume that Reviewer #2 has concerns on two different analyses, namely the GL approach discussed in Supplementary Note S7 and the multi-domain resistance circuit model discussed in Supplementary Note S10. Below, we explain that these two analyses are simple and qualitative enough to avoid overanalysis.

Firstly, the GL approach in Supplementary Note S7 is just intended to explain a possible driving force of the nematic domain-wall motion under uniaxial strain. We showed that the minor domains with nematicity (indicated by the large H_{c2} direction) pointing $\pm 60^\circ$ or $\pm 30^\circ$ from the external uniaxial strain direction are energetically unfavorable by using a very simple GL formalism. Here we only considered two terms, namely the coupling term

between the nematic superconductivity and the external strain, and the term for the coupling between the nematic superconductivity and the pre-existing symmetry breaking field. Even with such a minimal model, the observed nematic-domain control is qualitatively explained. Because we used the simplest GL formalism here and its conclusion is rather qualitative, this analysis should not be considered as “overanalysis”.

To emphasize the simplicity of the GL analysis, we modified the first paragraph in Discussion (page 9 of the Main Text) and the first paragraph in Supplementary Note S7.

Secondly, for the circuit model used in the resistivity simulation, we would like to emphasize that this model is not purely theoretical; we used experimental $R(H)$ curves (measured under high compressive strain) to model the behavior of each domain (See “Magnetoresistance of each domain” subsection of Supplementary Note S10). We did not need to do any fine tuning of the functions and parameters for the resistivity behavior near the superconducting transition, but we just simply fit the experimental data with an empirical function. Thus, our model is quite simple and robust, independent of the choice of resistivity mechanism and parameters. Then, by modeling a simple circuit consisting of such domains, we evaluated the resistive upper critical field. Here we used equations for classical circuit analysis, which is again independent of the choice of detailed mechanism of resistivity. From these considerations, we believe that our circuit model is simple enough to avoid overanalysis. We would like to state that we tried several circuit models with various element configurations and confirmed that the results of the simulations are rather similar.

In order to elaborate on the simplicity, qualitative strength, and robustness of these analyses, we added a few sentences in pages 7-8 of the Main Text and in the beginning of Supplementary Note S10.

[Comment 4 of Reviewer #2]

Therefore, I would suggest a revision of the manuscript before the consideration for acceptance of the manuscript.

[Our response]

We thank again Reviewer #2 for careful review and positive evaluation. We now revised the manuscript by considering all of their comments. Now we believe that the manuscript is acceptable for publication in Nature Communications.

Reviewer #3

[Comment 0 of Reviewer #3]

In the paper by Kostylev et al. a detailed study of the strain dependence of the nematic superconductivity in $Sr_xBi_2Se_3$ is presented. The nematic superconductivity is monitored by the anisotropy in the upper critical field. This is a timely topic in the center of the field of topological superconductors and certainly of sufficiently broad scientific interest for Nature Communications. The paper is well written and the data and analysis are sound. The authors present the first control of superconductivity via uniaxial strain in doped Bi_2Se_3 superconductors, in that sense it is pioneering work. The experiments are properly described and the outcome is clear. The angular variation of the resistance is compared with a simulation to provide a further understanding of the data.

The quality of the paper is high and I expect it to have a significant impact. I recommend publication in Nature Communications. Below are a few minor comments that should be addressed prior to publication.

[Our response]

We thank Reviewer #3 for their careful review of our manuscript. We are pleased that they highly evaluated our study by stating “*The quality of the paper is high and I expect it to have a significant impact*” and recommended publication in Nature Communications. Below, we respond to their comments, which are quite helpful to improve

our manuscript. For your convenience, we submit another manuscript file, in which revisions are indicated in red.

[Comment 1 of Reviewer #3]

1. On page 1 line 6 "are" should read "is" and on page 2 line 25 "have" should read "has".

[Our response]

We thank Reviewer #3 for careful reading. These typos are now fixed.

[Comment 2 of Reviewer #3]

2. On page 3, line 42 the sample to sample variation is discussed. Here a reference to the paper by Kuntsevich in New Journal of Physics (20-2018-103022) is missing.

[Our response]

In the previous version of the manuscript, several important references were missing because of the limits in the text length and the number of references of a journal where the manuscript was initially submitted. For Nature Communications, we can put more text and references. Following this comment, we now cited Kuntsevich *et al.* [New J. Phys. 2018] in page 3 of the Main Text. We also included several new references in page 2. Moreover, we newly wrote several paragraphs in pages 11-12 describing and discussing some of these newly added references.

[Comment 3 of Reviewer #3]

3. Kuntsevich et al. also published strain dependence in SrxBi2Se3 probed by the combination of XRD and transport. the authors should make a link and comparison to this work (Phys. Rev. B 100-20190224509).

[Our response]

We now also cited Kuntsevich *et al.* [Phys. Rev. B 2019] in page 2, as well as in the new paragraph added in pages 11-12 of the Main Text, with explaining comparison with our work.

[Comment 4 of Reviewer #3]

4. I am puzzled by the change of Tc in Fig.S3 when the current is decreased from 250 to 50 microA. Normally such differences can be attributed to Joule heating of the sample for the larger current (the sample then has a higher temperature than the thermometer). The authors should explain whether local heating can be excluded. Overall the current in all experiments is large (250 microA) and absence of Joule heating should be demonstrated.

[Our response]

We thank Reviewer #3 for pointing out this issue of Joule heating. Our conclusion is: (1) Joule heating is totally negligible in the results obtained in the elastic regime and (2) it is very probably insignificant even in the data in Fig.S3 obtained in the plastic regime, as discussed below.

Firstly, on the claim (2), the data in Fig.S3 was measured after applying a very large compressive strain (-2.18%), exceeding the elastic limit. This strain very probably caused cracks in the sample, as evidenced by the increase in the normal-state resistance by 2 m Ω and by the absence of zero-resistivity even in the superconducting state. This increase of resistance can cause an additional Joule heating of 0.005 nW for $I = 50 \mu\text{A}$ and 0.125 nW for $I = 250 \mu\text{A}$.

Such heating power is quite tiny. To demonstrate that, we can compare this heating power with a commonly-used rule of thumb for low-temperature thermometry. This rule tells that the power input to a resistive thermometer at temperature T should be carefully reduced below the upper limit $P_{\text{ul}} = T^3$ to avoid self-heating of the thermometer, where P_{ul} is in the unit of nW and T is in K. Joule heating in a sample should be also reduced below P_{ul} . In the present case, this rule yields P_{ul} of 22 nW at 2.8 K, which is much higher than the heating that we are now concerned.

If the current dependence of T_c ($T_c \sim 2.88$ K for $I = 50$ μA and ~ 2.4 K for $I = 250$ μA ; Fig.S3) is solely due to Joule heating, the sample temperature must be increased by 0.48 K with the 0.12-nW heating power. To achieve such a large temperature increase with a tiny heating power, the thermal conductance between the sample and the thermometer must be as small as 0.25 nW/K. Considering the situation that there are multiple and strong thermal connections between the sample and thermometer, through the four gold wires for resistivity measurements or through the uniaxial-strain device that are made of titanium, such a small thermal conductance is unrealistic. Notice that gold and titanium are both good thermal conductors even at 1.8 K. Just for a demonstration, gold has very high thermal conductivity of ranging 1-10 W/(K cm) at around 3 K depending on the quality (G. K. White, *Proc. Phys. Soc. A* **66** 559 (1953).) and one needs a 60-600 meter long wire even if the diameter is as small as 20 μm (as the wire we used in our experiment) to achieve 0.25 nW/K.

Secondly, on the claim (1), we have checked that the critical temperature or fields do not depend on the applied current up to 250 μA for the data other than those in Fig.S3 (i.e. all data measured BEFORE applying strain exceeding the elastic limit). Please see Fig. R2 below (also added as Fig. S16 in the Supplementary Information); the data measured with 100 and 250 μA almost coincide each other, although there is some suppression of superconductivity with 400 μA . Thus, for the data presented in the main text, Joule heating or any other current-induced effect should be negligible.

We agree that the possibility of Joule heating is very important. To describe the claim (1) to the readers, we added a sentence in the “Resistivity measurement” subsection of Methods, and a new subsection in the Supplementary Information (Supplementary Note S12) with a new figure (Fig. S16). For the claim (2), we added one paragraph in Supplementary Note S3.

Figure R2: Comparison of resistance measured with various currents. The left panel shows the temperature dependence measured at zero field, and the right panel shows the field sweep measured at 2.2 K and for fields nearly parallel to the x axis (one of the a axes, parallel to the strain direction). For both panels, the data measured with 100 and 250 μA almost coincide each other, although there is some suppression of superconductivity with 400 μA .

We thank Reviewer #3 again for careful reading and positive feedback. We have revised the manuscript by taking into account all of their comments. We now believe that the revised manuscript is acceptable for publication in Nature Communications.

REVIEWER COMMENTS

Reviewer #1 (Remarks to the Author):

Unfortunately authors seem to have not understood my comments regarding H_{c2} . The standard definition of H_{c2} is the field at which superconductivity onsets, at a given temperature. For a uniform sample, the order parameter amplitude at this point is infinitesimal, so there can be no pinning whatsoever.

If sample is spatially inhomogeneous, then different parts will nucleate at different temperatures. Then, at a given temperature different parts will have different H_{c2} 's. That would make the whole picture much more complicated, and there could be "anisotropies" in " H_{c2} " related to the shape of domains, not only type of OP within them.

In this sense, the "standard" definition of H_{c2} is much more useful in my opinion (assuming sample is sufficiently uniform), since only symmetries of the system determine the anisotropes, and pinning is irrelevant. If sample is not very uniform, it is better to focus on the higher values of R/R_n . The reason for this is that in this regime the H_{c2} measurement only picks up the most robust (in the sense of superconductivity) regions of the sample and thus the effective uniformity is enhanced (the less superconducting regions are already normal by then, and do not complicate the analysis). Indeed, authors say that this is where the 6fold symmetry is the most pronounced.

On the other hand, at a weaker H field, the story is more complicated, since some regions are more superconducting, some less, in a realistic sample. There, the network model as authors propose may be relevant, but as I mentioned above, the shape of domains will matter there as well, and analysis will be in general much more tricky in my opinion.

To summarize, I am still not convinced by the modeling that authors do. I would recommend that they try to extract the best approximation of H_{c2} as they can and focus on that in their analysis. If they think I am wrong in my understanding -- I'd like to know why.

Reviewer #2 (Remarks to the Author):

The revisions addressed my questions. I am glad that I helped in the process.

Response to Reviewer #1's comments

To Reviewer #1

First of all, we would like to thank the Reviewer #1 for helping us to clarify any ambiguities in the interpretation of our results. Below, we list our point-by-point responses to your comments.

[Comment 1 of Reviewer #1]

Unfortunately authors seem to have not understood my comments regarding H_{c2} . The standard definition of H_{c2} is the field at which superconductivity onsets, at a given temperature. For a uniform sample, the order parameter amplitude at this point is infinitesimal, so there can be no pinning whatsoever.

[Our response]

We think that we have understood your previous comments, but our wording may not have been sufficiently clear enough in the previous response.

Firstly, we fully agree with your standard (but theoretical) definition of H_{c2} . Secondly, we seem to share another aspect that defining H_{c2} from experimental data is not always straightforward, as you point out in the following comments.

We also agree that, for *an ideal sample possessing complete three-fold rotational symmetry*, the nematicity of the superconducting order parameter is solely determined by the applied magnetic field direction. However, in reality, each sample of doped Bi_2Se_3 seems to have a pre-existing symmetry breaking field (SBF), which can be attributable to small in-plane strain, uniaxial arrangements of doped atoms, or small monoclinic lattice distortions discussed in a recent preprint [T. Frohlich *et al.*, arXiv:2004.10553]. As discussed in the theory paper [Ref. 34; J. W. F. Venderbos *et al.*, *Phys. Rev. B* **94**, 094522 (2016).], under such a realistic situation where a finite pre-existing SBF exists, the in-plane H_{c2} should exhibit predominantly two-fold anisotropy (See Fig.2 of Ref. 34). This is the situation that we are considering, and is indeed observed in many other experimental works. In order to clarify these considerations we modified a paragraph in the main text (pg. 7).

[Comment 2 of Reviewer #1]

If sample is spatially inhomogeneous, then different parts will nucleate at different temperatures. Then, at a given temperature different parts will have different H_{c2} 's. That would make the whole picture much

more complicated, and there could be "anisotropies" in "Hc2" related to the shape of domains, not only type of OP within them.

[Our response]

Yes, we appreciate that you understand that we have to exercise due caution when defining H_{c2} from experimental data. Roughly speaking, the measured H_{c2} with low R/R_n criteria represents an average of properties within the sample if there's some inhomogeneity (such as domains, T_c distribution, etc.). Nevertheless, as you mentioned, the measured H_{c2} can be additionally affected by various factors such as connectivity among domains, domain shape, etc. Revisions related to this comment are described in the following responses.

[Comment 3 of Reviewer #1]

In this sense, the "standard" definition of Hc2 is much more useful in my opinion (assuming sample is sufficiently uniform), since only symmetries of the system determine the anisotropes, and pinning is irrelevant. If sample is not very uniform, it is better for focus on the higher values of R/Rn. The reason for this is that in this regime the Hc2 measurement only picks up the most robust (in the sense of superconductivity) regions of the sample and thus the effective uniformity is enhanced (the less superconducting regions are already normal by then, and do not complicate the analysis). Indeed, authors say that this is where the 6fold symmetry is the most pronounced.

[Our response]

Yes, we agree with your picture. For a realistic situation with certain non-uniformity in the sample, H_{c2} defined using higher R/R_n values should be more useful to understand what is happening in the sample. In our manuscript, indeed, we have already put more focus on H_{c2} defined by $R/R_n = 95\%$ than those by lower criteria. Following your advice we added one sentence in the main text (pg. 8) to emphasize that with the definition using higher R/R_n we can pick up the most robust part of the domain structure.

Moreover, following your suggestion, we made additional analyses on this onset H_{c2} , as shown below. The result in Fig. R1 demonstrates that, although the 6-fold component is most pronounced for H_{c2} using 95% criteria, the experimental H_{c2} data is impossible to be fitted using the simple 6-fold formula $H_{c2}(\varphi_{ab}) = H_0 + H_6 \cos(6\varphi_{ab})$ (orange curve in Fig. R1) that is suggested theoretically for the ideal no-pinning situation [Ref. 34; J. W. F. Venderbos *et al.*, *Phys. Rev. B* **94**, 094522 (2016)]. The dominant two-fold behavior is considerably large. We also tried to fit the data with our multi-domain model and the fitting was reasonably successful as shown by the blue curve in Fig. R1. Note how the multi-domain model accounts for the difference of the peak widths between the domains.

Figure R1: In-plane field-angle ϕ_{ab} dependence of H_{c2} defined with the $R/R_n = 95\%$ criteria for zero applied strain ($\Delta\varepsilon_{xx} = 0$). The black points are the measured data and the blue curve is the fitting based on the multi-domain nematic model explained in the text. The orange curve is the result of a trial fitting by using a six-fold sinusoidal function $H_{c2}(\phi_{ab}) = H_0 + H_6 \cos(6\phi_{ab})$ expected for the ideal situation with the absence of pre-existing symmetry breaking field [Ref.34]. This result demonstrates that the simple six-fold behavior alone cannot explain the data.

Additionally, we find that with increasing compressive strain there is a smooth and anisotropic disappearance of the 6-fold pattern. In particular, the pair of “satellite peaks”: -150° & $+30^\circ$ and -30° & $+150^\circ$ have a different strain dependence; the former pair is still evident even under -1.19% strain whereas the latter pair completely disappears (see Fig. R2 below). If the 6-fold component of H_{c2} is indeed due to a single 6-fold domain, then with the application of strain H_{c2} will suddenly exhibit a 2-fold behavior and both satellite peaks should be gone simultaneously. However, we do not observe this.

Figure R2: Same data as Fig. R1 but under high compressive strain ($\Delta\epsilon_{xx} = -1.19\%$). Notice that the shoulder peak at $\varphi_{ab} = -30^\circ$ is completely suppressed, whereas the other shoulder peak at $\varphi_{ab} = -150^\circ$ is still visible even under this high compression. Indeed, the multi-domain model is able to account for the strain induced change.

These two facts are rather difficult to be explained by any single domain models, and urges us to consider multiple nematic domains. We added this explanation to the main text, and this analysis together with these two figures in Sec. S10 of the Supplementary Information.

In addition, during this discussion with you, we noticed that some minor domains can exhibit the ideal 6-fold behavior, if there is an accidental cancellation between the pre-existing SBF and the external strain. Thus, we added several sentences in the main text (pg. 7) to explain this additional interesting possibility.

[Comment 4 of Reviewer #1]

On the other hand, at a weaker H field, the story is more complicated, since some regions are more superconducting, some less, in a realistic sample. There, the network model as authors propose may be relevant, but as I mentioned above, the shape of domains will matter there as well, and analysis will be in general much more tricky in my opinion.

[Our response]

We agree that analyses of H_{c2} with lower R/R_n can be complicated; the detailed behavior can be determined by various factors such as domain shapes and configurations. Nevertheless, we believe that our simple network circuit model captures the *qualitative* behavior of our sample quite well, as we demonstrate below, in spite of the possible complications that arise in reality as discussed above.

In Fig. R3, we show results of simulation using various circuit models. Notice that the parameters used here are determined by the fitting using 3D model (1) to the data at 95%. Nevertheless, for the 95% H_{c2} , 3D model (2) gives very similar results as 3D model (1), and the results from the other two models still qualitatively matches the data. For the 5% H_{c2} , although the parameters optimized with the 95% data are used, we still observe a qualitatively consistent pattern (except for 3D model (3); as discussed below). From these facts, we believe that our network model indeed provides the simplest basis to explain all the observed phenomena including strain dependence.

Figure R3: Calculated in-plane upper critical field (defined by the 95 and 5% criteria) by the circuit model using various configurations of nematic domains illustrated in the bottom of the figure. Experimental data under zero applied strain are also plotted (black circles). For

the calculation, we used parameters obtained with the fitting in Fig. R1 for the standard 3D circuit model (3D model (1); orange curve).

Note that for the simulation using 3D model (3) at $R/R_n = 5\%$ (Fig. R3, right-panel, red dotted curve) does not reproduce the observed two-fold structure with peaks at $\varphi_{ab} = \pm 90$ deg. The reason is that it breaks a fundamental assumption in the way the model should be constructed; that is, that the current has to pass through at least one Y_0 domain when going from the left to right electrodes. This rule avoids a short circuit between the electrodes when both Y_1 and Y_2 domains superconduct. Therefore, as long as this important assumption is maintained, we can expect qualitatively good results despite the particularities of the circuit configuration in the model.

Moreover, from an experimental point of view, H_{c2} defined with other (i.e. lower R/R_n) criteria also has various important meanings. For example, H_{c2} defined with R/R_n close to zero is important since they are comparable to H_{c2} deduced from thermodynamic quantities such as specific heat or magnetization, and thus are quite important when considering consistency with other works. Therefore, we, as experimentalists, believe that showing H_{c2} of various criteria as well as their simple analyses are quite important.

[Comment 5 of Reviewer #1]

To summarize, I am still not convinced by the modeling that authors do. I would recommend that they try to extract the best approximation of H_{c2} as they can and focus on that in their analysis. If they think I am wrong in my understanding -- I'd like to know why.

[Our response]

As explained above, we added more analyses in this revised manuscript of the onset (95% criteria) H_{c2} , which reveals further confirmation for the necessity to consider a multi-domain picture. We also mentioned in this revision the possibility that some of the domains may exhibit the ideal 6-fold behavior without pinning. For the modeling, we showed simulations obtained using other network-circuit configurations, to demonstrate the effectiveness of the simulation of the network model.

Although you recommend to “focus on” the onset H_{c2} , we believe that deleting H_{c2} of other criteria is not a reasonable approach, considering the aspect that these other H_{c2} characterizations have independent important meanings as we explained above. Thus, we would like to keep them.

To summarize, we revised the manuscript by fully considering your comments. We hope that you are satisfied by these responses and revisions.